# Chromatin accessibility variation provides insights into missing regulation underlying immune-mediated diseases

Raehoon Jeong[1,2], Martha L Bulyk[1,2,3]*

[1]Division of Genetics, Department of Medicine, Brigham and Women's Hospital and Harvard Medical School, Boston, United States; [2]Bioinformatics and Integrative Genomics Graduate Program, Harvard University, Cambridge, United States; [3]Department of Pathology, Brigham and Women's Hospital and Harvard Medical School, Boston, United States

## eLife Assessment

This paper addresses a significant question regarding the low overlap between genetic discoveries for human complex diseases and those for gene expression by emphasizing the contribution of cell-type-specific chromatin accessibility QTLs. The analyses supporting the main claims are **convincing**, and the key conclusions are **valuable** and of interest to readers in the fields of human genetics and functional genomics.

*For correspondence:
mlbulyk@genetics.med.harvard.edu

Competing interest: The authors declare that no competing interests exist.

**Abstract** Most genetic loci associated with complex traits and diseases through genome-wide association studies (GWAS) are noncoding, suggesting that the causal variants likely have gene regulatory effects. However, only a small number of loci have been linked to expression quantitative trait loci (eQTLs) detected currently. To better understand the potential reasons for many trait-associated loci lacking eQTL colocalization, we investigated whether chromatin accessibility QTLs (caQTLs) in lymphoblastoid cell lines (LCLs) explain immune-mediated disease associations that eQTLs in LCLs did not. The power to detect caQTLs was greater than that of eQTLs and was less affected by the distance from the transcription start site of the associated gene. Meta-analyzing LCL eQTL data to increase the sample size to over a thousand led to additional loci with eQTL colocalization, demonstrating that insufficient statistical power is still likely to be a factor. Moreover, further eQTL colocalization loci were uncovered by surveying eQTLs of other immune cell types. Altogether, insufficient power and context specificity of eQTLs both contribute to the 'missing regulation'.

## Introduction

More than a decade of genome-wide association studies (GWAS) has revealed several properties of the genetic architecture of complex traits and diseases (*Visscher et al., 2017*; *Claussnitzer et al., 2020*). Most (~93%) of the genetic associations are detected in the noncoding portion of the genome (*Maurano et al., 2012*), and disease heritability is concentrated in putative regulatory regions (*Gusev et al., 2014*). Expression quantitative trait loci (eQTLs), which are loci associated with gene expression levels, are enriched for trait associations (*Nicolae et al., 2010*; *Hormozdiari et al., 2018*). Complex traits are also characterized by their extreme polygenicity, where individual genetic association has only a small effect on the trait (*O'Connor et al., 2019*). Altogether, these observations have led to a prevalent theory that causal genetic variants affect regulation of key genes across the genome, where each gene explains a modest proportion of trait variation (*Boyle et al., 2017*). There are experimental

strategies aimed at nominating putative causal genes at noncoding GWAS loci (*Nasser et al., 2021*; *Weeks et al., 2023*; *Morris et al., 2023*). As an alternate approach, an eQTL signal colocalizing with the GWAS signal illustrates the effect of the causal variant on gene expression and suggests that the affected gene contributes to the trait. Detection of disease-associated eQTLs thus can identify putative disease genes, helping to elucidate disease mechanisms and develop therapeutics targeting them (*Plenge et al., 2013*).

Although it has become expected that eQTLs will be discovered in most noncoding GWAS loci, only a minority of trait-associated loci have been explained by eQTLs (*Chun et al., 2017*; *Barbeira et al., 2021*; *Connally et al., 2022*; *Yao et al., 2020*). The Genotype-Tissue Expression (GTEx) eQTL study across 49 human tissues recognized that, for a typical complex trait, about 20% of GWAS loci contained a colocalized eQTL in the *cis* region (i.e. 1 Mb) around the gene (*i.e. cis*-eQTL) (*Barbeira et al., 2021*). Even when focusing just on putatively causal genes, the rate of colocalization was very low (8%) (*Connally et al., 2022*). Furthermore, the proportion of trait heritability mediated by *cis*-eQTLs ($h^2_{med}/h^2_{SNP}$) of assayed gene expression was estimated to be only about 11% on average (*Yao et al., 2020*). We will call the missing link between genetic association to traits and regulatory function of the associated noncoding variants as 'missing regulation', as *Connally et al., 2022*, introduced. To be able to detect eQTLs in the unexplained disease-associated loci, a better understanding of the possible reasons for why they have been missing is essential.

There are many possible explanations for why disease-associated loci are missing colocalized eQTLs (*Connally et al., 2022*; *Umans et al., 2021*; *Mostafavi et al., 2023*; *Hukku et al., 2021*) and for why $h^2_{med}/h^2_{SNP}$ estimates for eQTLs are relatively low (*Yao et al., 2020*). First, statistical power to detect disease-associated eQTLs may be insufficient (*Hukku et al., 2021*). For example, negative selection against gene expression variation may lead to challenges in detecting eQTLs for trait-relevant genes (*Mostafavi et al., 2023*; *Glassberg et al., 2019*). Since disease genes are likely to be dosage sensitive, there would be selection against their having large eQTL effects (*Glassberg et al., 2019*). Consequently, the negative selection induces a 'flattening' effect, in which weak eQTL variants, often in regions distal to the gene's promoter (*Dimas et al., 2009*), may reach high enough frequency, whereas strong eQTL variants remain at low frequency (*O'Connor et al., 2019*). In fact, eQTL-mediated heritability was enriched in genes showing mutational constraint and those with lower *cis*-heritability (*Yao et al., 2020*). These weaker or low-allele-frequency eQTL effects would require larger sample sizes to be detected with statistical significance and to show colocalization (*Hukku et al., 2021*). Second, causal eQTL effects may be specific to cell types that have not been assayed (*Umans et al., 2021*). For example, immune cell eQTLs are highly cell-type specific, and eQTL effects specific to some immune cell types may mediate immune disease risk (*Schmiedel et al., 2018*). Specificity of eQTL effects can also be limited to specific cell states (*Alasoo et al., 2018*; *Nathan et al., 2022*). Detecting cell-type or cell-state-specific eQTL effects requires the necessary gene expression datasets from the relevant cell types and states (*Umans et al., 2021*), which has been a limiting resource for such analyses.

To investigate why disease-associated eQTL signals have been missing, we focused on immune-mediated diseases (IMDs) as a model set of complex traits. We aimed to collect IMD-associated loci that are expected to show eQTL signals in some cell type. Since active regulatory elements coordinate target gene expression (*Field and Adelman, 2020*), we reasoned that variants that affect chromatin phenotypes at regulatory elements, such as transcription factor (TF) binding (*Kasowski et al., 2010*; *Kilpinen et al., 2013*; *Waszak et al., 2015*) and chromatin accessibility (*Degner et al., 2012*; *Kumasaka et al., 2019*), have the potential to impact gene expression (*Albert and Kruglyak, 2015*). These chromatin phenotypes may show detectable genetic effects even when an eQTL effect in the same cell type was not identified in the locus (*Wu et al., 2023*). For example, only about 20% of lymphoblastoid cell lines' (LCLs') PU.1 binding QTLs (bQTLs) that colocalized with blood cell traits' association showed an eQTL effect for a nearby gene in LCLs (*Jeong and Bulyk, 2023*).

Here, we analyzed genetic and functional genomic (i.e. ATAC-seq and RNA-seq) data in LCLs. LCLs are derived from B lymphocytes, and their *cis*-regulatory elements were enriched for variants associated with some IMDs (*Kundaje et al., 2015*; *Farh et al., 2015*). We evaluated whether chromatin accessibility QTLs (caQTLs) in LCLs potentially explain IMD associations using mediated heritability analysis (*Yao et al., 2020*) and colocalization (*Giambartolomei et al., 2014*; *Pickrell et al., 2016*). Then, we searched for disease-associated loci that were significant caQTLs, but not eQTLs.

We examined whether the various potential reasons for missing eQTLs can account for IMD-associated loci that are explained by caQTLs but not eQTLs. First, we explored the extent to which eQTLs may have been missed because of limited statistical power. We compared *cis*-heritability of colocalized caQTLs and eQTLs stratified by distance between the accessible region and the transcription start site (TSS) of the associated gene. We also investigated whether meta-analysis of published LCL eQTL summary statistics, in order to effectively increase the sample size, can uncover previously missed eQTLs. Second, we surveyed whether cell-type specificity of regulatory variant effect may account for the missing regulation. We surveyed various immune cell eQTL data to identify loci with which they colocalize even if LCL eQTLs did not colocalize with those loci.

Through this study, we present how regulatory QTLs beyond eQTLs, such as caQTLs, can be effective in detecting the potential molecular consequences of disease-associated variants. Moreover, results from inspecting disease-associated loci where genetic effects are detected on chromatin accessibility but not on expression suggest reasons why the effects on gene expression may have been missed. These results provide insights on which strategies may be effective in uncovering more genes that underlie diseases.

## Results

### Accessible chromatin in LCLs explains a significant proportion of immune-mediated disease heritability

We aimed to evaluate whether variants that alter chromatin accessibility in LCLs may explain genetic associations to IMD. First, we verified whether accessible regions in LCLs are enriched for IMD heritability. We reanalyzed 100 LCL ATAC-seq samples (*Kumasaka et al., 2019*) to define accessible regions in this cell type. With stratified LD score regression (S-LDSC) (*Finucane et al., 2015*), we estimated their heritability enrichment across 13 IMDs, including 11 autoimmune diseases – autoimmune thyroid disease (ATD) (*Cordell et al., 2021*), celiac disease (CEL) (*Dubois et al., 2010*), Crohn's disease (CD) (*de Lange et al., 2017*), inflammatory bowel disease (IBD) (*de Lange et al., 2017*), juvenile idiopathic arthritis (JIA) (*López-Isac et al., 2021*), multiple sclerosis (MS) (*Patsopoulos, 2019*; https://imsgc. net/), primary biliary cholangitis (PBC) (*Cordell et al., 2021*), rheumatoid arthritis (RA) (*Ishigaki et al., 2022*), systemic lupus erythematosus (SLE) (*Bentham et al., 2015*), ulcerative colitis (UC) (*de Lange et al., 2017*), and vitiligo (VIT) (*Jin et al., 2016*) – and 2 allergic diseases – allergy (ALL) (*Loh et al., 2018*) and asthma (AST) (*Loh et al., 2018*). We also analyzed genome-wide association study (GWAS) data for 3 non-immune diseases – type 2 diabetes (T2D) (*Morris et al., 2012*), coronary artery disease (CAD) (*Schunkert et al., 2011*), and schizophrenia (SCZ) (*Schizophrenia Working Group of the Psychiatric Genomics Consortium, 2014*) – for comparison. Single nucleotide polymorphisms (SNPs) in accessible regions in LCLs were significantly enriched for IMD heritability (p<0.003125 [Bonferroni-corrected threshold], S-LDSC; *Figure 1A*) and there was no significant enrichment for nonimmune diseases (p>0.05, S-LDSC). These results indicate that accessible regions in LCLs harbor many variants specifically associated with IMDs, and therefore that LCLs share IMD-associated accessible regions with those of the causal cell type(s).

Next, we applied mediated expression score regression (MESC) (*Yao et al., 2020*) to investigate the causal relationship between caQTLs in LCLs and IMD associations (*Figure 1—figure supplement 1A*). Compared to S-LDSC analysis that tests for heritability enrichment of SNPs with some functional annotation (e.g. accessible regions), MESC analysis specifically estimates the heritability that is mediated (i.e. $h^2_{med}$) by the SNPs' *cis*-effects on a molecular phenotype (e.g. caQTLs). We estimated that caQTLs in LCLs mediate 16.3–42.7% of autoimmune disease heritability and 8.5–9.4% of allergic disease heritability (*Figure 1B*). For nonimmune diseases, the estimates were lower and not significant (p>0.003125 [Bonferroni-corrected threshold], MESC). Interestingly, SCZ showed a nominally significant proportion of caQTL-mediated heritability in LCLs (p<0.05, MESC), consistent with the hypothesis that B cells may play some role in SCZ pathogenesis (*Schizophrenia Working Group of the Psychiatric Genomics Consortium, 2014*; *van Mierlo et al., 2019*). Our results indicate that LCLs are a valid cell type in which to search for caQTLs that mediate genetic risk for 7 IMDs – CD, IBD, MS, PBC, RA, SLE, and UC – but not for allergic diseases. In subsequent analyses, we focused on 7 IMDs – CD, IBD, MS, PBC, RA, SLE, and UC – that showed significant caQTL-mediated heritability (p<0.003125 [Bonferroni-corrected threshold], MESC).

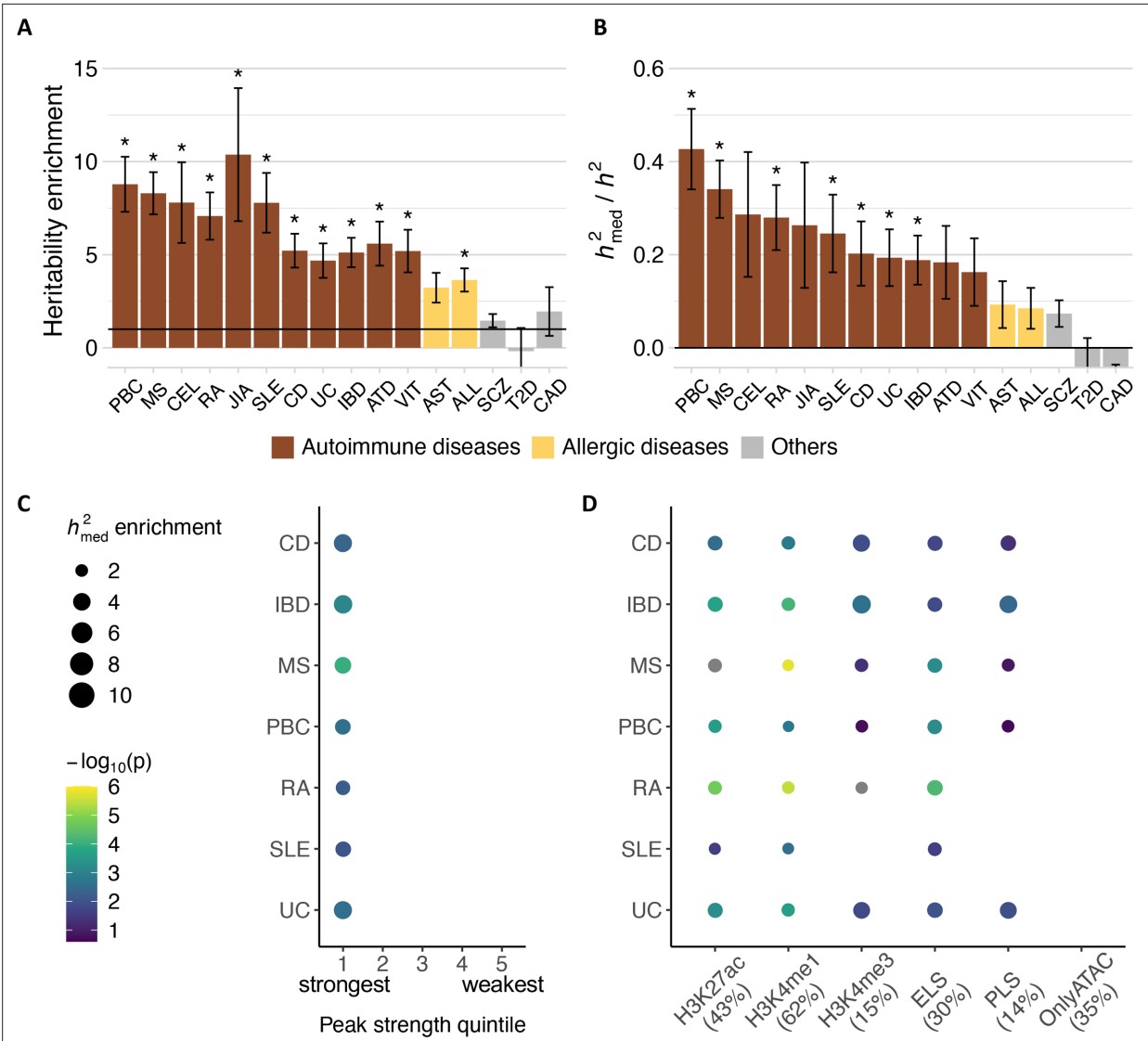

**Figure 1.** Immune disease heritability mediated by chromatin accessibility in lymphoblastoid cell lines (LCLs). (**A–B**) Heritability enrichment in accessible regions in LCLs, based on (**A**) stratified linkage disequilibrium (LD) score regression (S-LDSC) and (**B**) the proportion of heritability mediated by chromatin accessibility quantitative trait loci (caQTLs) in LCLs based on mediated expression score regression (MESC). For both, error bars represent jackknife standard errors of the mean. The color of the bars indicates disease type. *: p<0.003125 (Bonferroni-corrected). (**C**) Mediated heritability enrichment of accessible regions by peak strength quintile. The strongest peaks are in the 1st quintile, and the weakest peaks are in the 5th quintile. Only enrichment values with FDR < 5% (based on q-value) are shown. Stronger and more significant enrichment indicates that mediated heritability is concentrated in that subset. (**D**) Mediated heritability enrichment of accessible regions by histone mark annotation. The percentages in parentheses represent the proportion of accessible regions with the indicated histone mark. Only enrichment values with FDR < 5% (based on q-value) are shown. Color and size of the points are on the same scale as in (**C**). PBC, primary biliary cholangitis; MS, multiple sclerosis; CEL, celiac disease; RA, rheumatoid arthritis; JIA, juvenile idiopathic arthritis; SLE, systemic lupus erythematosus; CD, Crohn's disease; UC, ulcerative colitis; IBD, inflammatory bowel disease; ATD, autoimmune thyroid disease; VIT, vitiligo; AST, asthma; ALL, allergies; SCZ, schizophrenia; T2D, type 2 diabetes; CAD, coronary artery disease.

The online version of this article includes the following figure supplement(s) for figure 1:

**Figure supplement 1.** Analysis workflow in this study.

**Figure supplement 2.** Proportion of chromatin accessibility quantitative trait locus (caQTL)-mediated immune-mediated disease (IMD) heritability explained by ATAC peaks with various histone marks.

**Figure supplement 3.** Properties of ATAC peaks with various histone marks.

## Regions with higher levels of accessibility and active histone marks explain most of caQTL-mediated heritability

To understand which features characterize accessible regions that mediate IMD heritability, we estimated $h^2_{med}$ enrichment (proportion of $h^2_{med}$/proportion of peaks) (*Yao et al., 2020*) in specific sets of accessible regions. We found that peaks with a larger number of nonredundant sequencing reads (i.e., 'stronger' peaks) in LCLs showed stronger $h^2_{med}$ enrichment (*Figure 1C*) and thus likely affect IMD-relevant gene expression more than 'weaker' peaks do. This observation is consistent with the 'Activity-by-Contact' model (*Nasser et al., 2021*), in which peaks with greater chromatin accessibility and H3K27ac ChIP-seq signal are predicted to have proportional effects on target gene expression.

Next, we considered peaks with the active histone marks H3K27ac, H3K4me1, or H3K4me3 (*Waszak et al., 2015*; *Delaneau et al., 2019*). Consistent with prior observations that putative cell-type-specific regulatory elements marked with H3K27ac and H3K4me1 are enriched for relevant disease associations (*Kundaje et al., 2015*; *Farh et al., 2015*), we found that caQTLs with H3K27ac and H3K4me1 marks in LCLs were significantly enriched for mediated IMD heritability (q-value <0.05, MESC; *Figure 1D*). Strikingly, both peak sets explained almost all of caQTL-mediated IMD heritability (*Figure 1—figure supplement 2*). Peaks with H3K4me3 marks, representative of promoters (*Heintzman et al., 2007*), also showed significant $h^2_{med}$ enrichment for most IMDs (q-value <0.05, MESC). Peaks with promoter-like signatures (i.e. H3K27ac and H3K4me3) (*Abascal et al., 2020*) and those with enhancer-like signatures (i.e. H3K27ac, but no H3K4me3) (*Abascal et al., 2020*) were also enriched for all IMD heritability (q-value <0.05, MESC). Conversely, peaks without any of the three active histone marks were completely depleted of caQTL-mediated IMD heritability (*Figure 1—figure supplement 2*). These 'ATAC-only' peaks were shorter, weaker, and further away from the TSS compared to peaks with active histone marks (*Figure 1—figure supplement 3*). Altogether, these results indicate that peaks characterized as putatively active regulatory elements explain nearly all of caQTL-mediated IMD heritability.

## caQTLs share IMD heritability with eQTLs and explain more of IMD heritability than do eQTLs

The model that gene regulatory activity explains a significant fraction of noncoding genetic associations to IMDs is supported by our findings that caQTLs mediate a significant proportion of IMD heritability and that those with active histone marks show strong $h^2_{med}$ enrichment. This is in contrast with relatively low average $h^2_{med}/h^2_{SNP}$ estimates (~11%) previously having been observed for eQTLs across 48 human tissues in GTEx and various human traits (*Aguet et al., 2020*). To directly compare the proportion of IMD heritability mediated by caQTLs and eQTLs in the same cell type (i.e. LCLs), we additionally applied MESC to gene expression data from LCLs (i.e. Geuvadis data, *Figure 1—figure supplement 1B*; *Lappalainen et al., 2013*).

Across the seven autoimmune diseases, the estimated proportion of heritability mediated by eQTLs ($h^2_{med; eQTL}/h^2_{SNP}$) ranged from 9% to 22% (*Figure 2A*). For all seven diseases, we estimated that eQTLs mediated less heritability than did caQTLs, even though the caQTLs' smaller sample size would potentially bias the estimates toward zero (*Yao et al., 2020*). A possible explanation is that some IMD-associated regulatory variants may show detectable effects on chromatin accessibility, but not on gene expression, in LCLs at the current sample size (n=373); such loci may account for the missing regulation.

We anticipated that IMD-associated variants that affect gene expression in *cis* do so by modulating regulatory element activity. Therefore, we investigated whether eQTL-mediated IMD heritability is shared by caQTL-mediated signals (*Figure 2B*). We performed MESC on both caQTLs and eQTLs together to estimate the amount of IMD heritability mediated by both collectively (Mediated heritability estimation of QTLs). For the 7 IMDs, the combined $h^2_{med; caQTL \cup eQTL}/h^2_{SNP}$ was only slightly higher (2.2–9.0%) than the estimates for just caQTLs ($h^2_{med; just caQTL}/h^2_{SNP}$; *Figure 2A*), suggesting that approximately 56–82% of eQTL-mediated heritability is shared with caQTL-mediated heritability (i.e. $h^2_{med; caQTL \cap eQTL}/h^2_{med; eQTL}$; *Figure 2—figure supplement 1*). These estimates are consistent with substantial sharing of caQTL- and eQTL-mediated IMD heritability. Nevertheless, 9–27% of IMD heritability is explained just by caQTLs, while only 2–9% of IMD heritability is explained just by eQTLs.

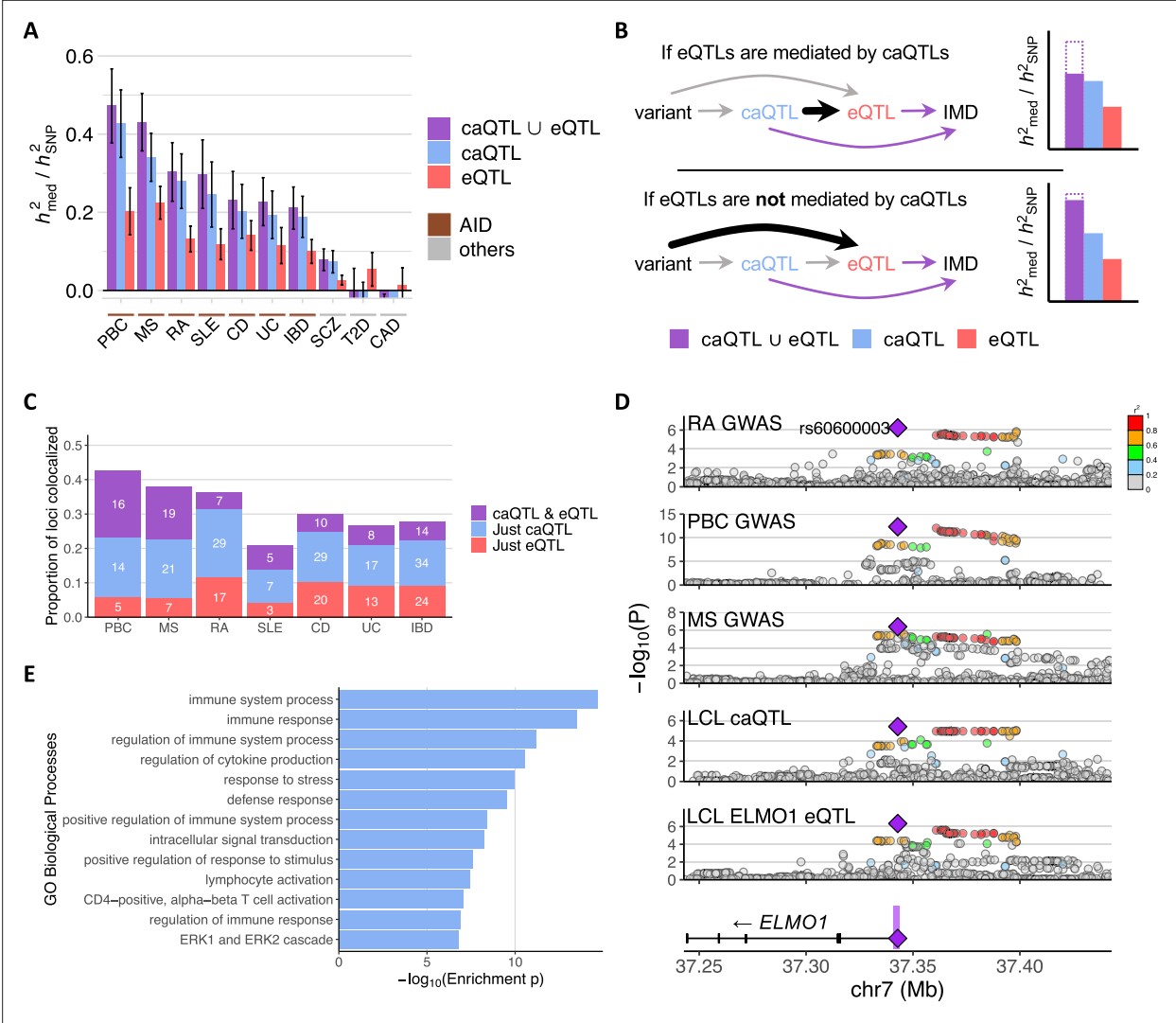

**Figure 2.** Immune-mediated diseases (IMD) heritability mediated by chromatin accessibility quantitative trait loci (caQTLs) and expression quantitative trait loci (eQTLs). (**A**) $h^2_{med}/h^2_{SNP}$ estimates of various IMDs for caQTLs, eQTLs, and their union. The error bars represent jackknife standard errors of the mean. AID: autoimmune disease. Disease abbreviations along the *x*-axis are as in *Figure 1*. (**B**) Schema of the potential causal relationships between genetic variants, caQTLs, eQTLs, and IMD risk. The two diagrams depict possible $h^2_{med}/h^2_{SNP}$ trends depending on the causal relationship between caQTLs and eQTLs. (**C**) Number of IMD-associated loci colocalized with caQTLs or eQTLs. The proportion is out of the total number of IMD-associated (p<10^-6) loci. (**D**) *ELMO1* locus plot showing association to rheumatoid arthritis (RA), primary biliary cholangitis (PBC), multiple sclerosis (MS), chromatin accessibility, and *ELMO1* expression in lymphoblastoid cell lines (LCLs). Purple shading in the gene plot at the bottom indicates the caQTL peak, and the purple diamond is the lead variant (rs60600003) that is within that peak. The other variants are colored by the degree of linkage disequilibrium (LD) with the annotated variant. (**E**) Enrichment of the Biological Process Gene Ontology (GO) terms of genes in proximity to IMD-colocalized caQTLs without eQTL colocalization.

The online version of this article includes the following figure supplement(s) for figure 2:

**Figure supplement 1.** Relationship between immune-mediated disease (IMD) heritability mediated by chromatin accessibility quantitative trait loci (caQTLs) and expression quantitative trait loci (eQTLs).

**Figure supplement 2.** Protein factors detected at colocalized chromatin accessibility quantitative trait locus (caQTL) by ChIP-seq.

**Figure supplement 3.** Biological processes enriched in immune-mediated disease (IMD)-colocalized expression quantitative trait loci (eQTLs).

**Figure supplement 4.** Immune-mediated disease (IMD) genome-wide association study (GWAS) colocalization with autoimmune disease drug target gene expression quantitative trait locus (eQTL) in lymphoblastoid cell lines (LCLs).

## Many IMD-associated loci show colocalization with caQTLs but not with eQTLs

We applied colocalization analysis (*Pickrell et al., 2016*) to identify IMD-associated loci that share genetic signals with caQTLs or eQTLs in LCLs. We selected candidate loci of 200 kb windows for each IMD with the following conditions: (1) lead IMD association at $p<10^{-6}$, (2) lead caQTL or eQTL association at $p<10^{-4}$, and (3) at least one variant simultaneously showed caQTL or eQTL $\chi^2$ statistics greater than 0.8×lead $\chi^2$ statistics for the caQTL or eQTL, respectively, and IMD association $\chi^2$ statistics greater than $0.8 \times \chi^2$ statistics for the IMD lead variant in the locus. We applied gwas-pw (*Pickrell et al., 2016*) and considered loci with posterior probability of colocalization (PPA3) >0.98 to be colocalized (*Kundu et al., 2022*). Some loci colocalized with only either a caQTL or an eQTL, while others colocalized with both (*Figure 2C* and *Supplementary file 1A and B*).

We investigated which proteins might be interacting with the colocalized caQTL peaks using Cistrome (*Liu et al., 2011*). We tested for overlap of the colocalized caQTL peak regions with ChIP-seq peaks detecting diverse proteins in immune-related cells (*Supplementary file 1C*). Consistent with the enriched mediated heritability in accessible regions with active histone marks (*Figure 1D* and *Figure 1—figure supplement 2*), proteins that are most often detected at the colocalized accessible regions are those related to RNA transcription (POL2RA and MED1), chromatin remodeling (EP300, BRD4, SMARCA4, and MTA2), or immune cell transcription factors (IKZF1, RUNX3, SPI1, RELA, RUNX1, and EBF1) (*Figure 2—figure supplement 2*). Interestingly, TRIM28, which functions as a repressor, was one of the most overlapping protein factors.

To confirm that the colocalized genes are relevant to IMD, we tested for their enrichment of Gene Ontology (GO) annotation terms for specific biological processes (*Thomas et al., 2022*). Considering all genes within 500 kb of the IMD GWAS lead variants at colocalized loci as background, the genes that showed eQTL colocalization for any IMD were enriched for various immune responses and signaling processes, such as 'positive regulation of immune system process' and 'regulation of lymphocyte activation' (*Figure 2—figure supplement 3*), indicating that the colocalized genes in LCLs are involved in immune function. For example, *IL6R* and *IL12A* encode direct or indirect targets of approved drugs – Tocilizumab and Ustekinumab – for autoimmune diseases like RA (*Sanmartí et al., 2018*) and CD (*Khanna and Feagan, 2013*). These two genes showed colocalization with both caQTLs and eQTLs in CD and PBC GWAS, respectively (*Figure 2—figure supplement 4A and B*). Increased *IL6R* expression was associated with higher risk for CD, and increased *IL12A* expression was associated with lower risk for PBC and SLE. The former observation is in line with Tocilizumab, a monoclonal antibody to IL-6 receptor, showing efficacy in CD patients (*Sanmartí et al., 2018*), although it is not pursued for approval because of potential side effects (*Monemi et al., 2016*). Interestingly, *IL6R* and *IL12A* eQTLs did not colocalize with the association signals of RA and CD, respectively, which are the diseases for which these drugs are approved (*Figure 2—figure supplement 4C and D*). Moreover, *ELMO1*, which previously had not been associated with autoimmune diseases, showed eQTL colocalization with RA, PBC, and MS association signals (*Figure 2D*). In all three, decreased *ELMO1* expression was associated with increased disease risk. In mice, *Elmo1* was required for polarization and migration of B and T lymphocytes (*Stevenson et al., 2014*).

Across the IMDs, there were many loci that colocalized with a caQTL but not with an eQTL (*Figure 2C*). These 'caQTL-only' loci showed enrichment for immune response genes in *cis* compared to all accessible regions in LCLs (*McLean et al., 2010*; *Tanigawa et al., 2022*), even though the colocalized eQTLs were enriched for immune response genes as well (*Figure 2E* and *Figure 2—figure supplement 3*), indicating that IMD-relevant genes without eQTL colocalization in Geuvadis LCL data (*Lappalainen et al., 2013*) are likely found in these loci.

## Distance to TSSs affects eQTLs but not caQTLs

Why might there be loci with caQTL colocalization only, despite the caQTL data having fewer samples than the eQTL data (100 vs 373)? Limited statistical power can prevent some eQTLs from being detected and showing significant colocalization (*Hukku et al., 2021*). As hypothesized by Mostafavi and colleagues, disease-relevant eQTLs may be weaker and more distal (*Mostafavi et al., 2023*). To understand the extent to which this effect may result in many loci showing colocalization only with caQTLs, we compared the *cis*-heritability ($h^2_{cis}$) of caQTLs and eQTLs depending on the distance from

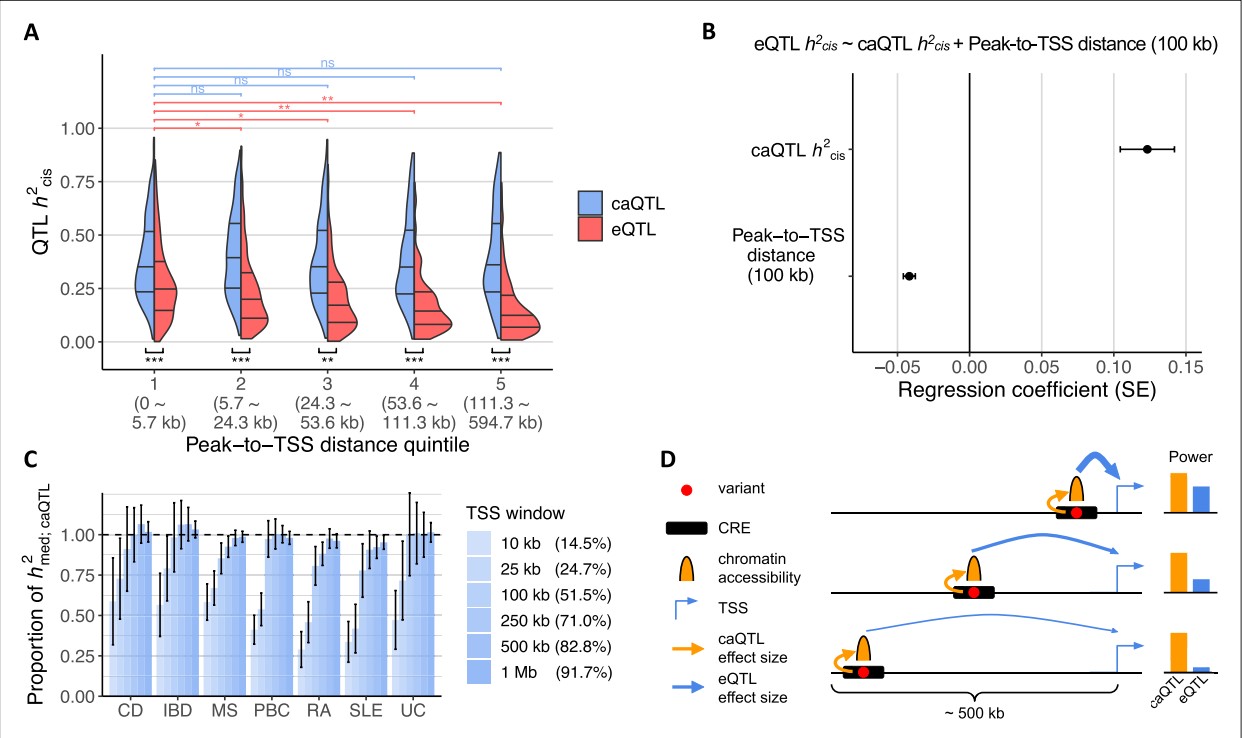

**Figure 3.** Effect of peak-to-TSS distance on *cis*-heritability of chromatin accessibility quantitative trait loci (caQTLs) and expression quantitative trait loci (eQTLs) and immune-mediated disease (IMD) heritability. (**A**) Distribution of *cis*-heritability ($h^2_{cis}$) of caQTLs and eQTLs by peak-to-TSS distance quintiles. The ranges of peak-to-TSS distance are shown in parentheses. The comparisons shown on the top (in respective colors) are between the nearest and each subsequent quintile of the respective QTL $h^2_{cis}$ distribution (i.e. one-sided Wilcoxon rank-sum test). The comparisons shown on the bottom (in black) are between caQTL and eQTL $h^2_{cis}$ distribution (one-sided paired Wilcoxon rank-sum test). *: $p<10^{-4}$, **: $p<10^{-10}$, ***: $p<10^{-20}$, and ns: $p>0.05$. (**B**) Regression estimates and their standard errors of the linear regression model testing the effects of caQTL $h^2_{cis}$ and peak-to-TSS distance on eQTL $h^2_{cis}$. Peak-to-TSS distance was expressed in units of 100 kb to neatly visualize the effect size estimates. The error bars represent standard errors of the regression estimate. SE: standard error. (**C**) Proportion of caQTL-mediated IMD heritability explained by ATAC peaks within various TSS windows. Percentage for each TSS window denotes the proportion of ATAC peaks in that window. The error bars represent jackknife standard errors of the mean. Disease abbreviations along the *x*-axis are as in *Figure 1*. (**D**) A model of the relationship between peak-to-TSS distance and power to detect a corresponding caQTL or eQTL. The thickness of the arrows indicates the variant effect size on chromatin accessibility (yellow) or gene expression (blue). TSS, transcription start site; CRE, *cis*-regulatory element.

the ATAC peak to the TSS of the gene (i.e. peak-to-TSS distance). We considered all caQTLs and eQTLs regardless of disease association.

We identified pairs of caQTLs and eQTLs that colocalized with each other (*Pickrell et al., 2016*), which implies that the regulatory variant modulating chromatin accessibility also affects gene expression. Then, the distance between the ATAC peak and the TSS of the eQTL gene (i.e. eGene) is the distance between a regulatory element and its target gene's TSS. We stratified the pairs into peak-to-TSS distance quintiles and compared the eQTL $h^2_{cis}$ distribution of the first quintile (i.e. closest pairs) with that of the later quintiles. We observed that eQTL $h^2_{cis}$ distribution decreased with increasing distance of the paired ATAC peaks from the TSS (p=1.0 × $10^{-4}$, 2.1×$10^{-10}$, 1.6×$10^{-15}$, and 1.7×$10^{-20}$, respectively, one-sided Wilcoxon rank-sum test; *Figure 3A*), consistent with the negative relationship between promoter-enhancer genomic distance and impact on gene expression (*Fulco et al., 2019*; *Zuin et al., 2022*). This result also explains why discovered eQTLs are concentrated near the promoter, where the variants are more likely to show stronger effects (*Võsa et al., 2021*). In contrast, caQTL $h^2_{cis}$ distribution was similar across peak-to-TSS distances (p>0.05, one-sided Wilcoxon rank-sum test; *Figure 3A*). For all distance quintiles, caQTL $h^2_{cis}$ was significantly higher than that of the paired eQTLs (p=4.0 × $10^{-23}$, 1.3×$10^{-24}$, 1.5×$10^{-19}$, 1.1×$10^{-28}$, and 2.4×$10^{-49}$, respectively, one-sided paired Wilcoxon rank-sum test; *Figure 3A*), and the contrast between them was greater at more distant quintiles, suggesting that the statistical power to detect and colocalize eQTLs is increasingly lower than that for caQTLs for regulatory effects far from the TSS.

Overall, caQTL $h^2_{cis}$ had a significant positive effect (p=6.6 × 10⁻¹¹, linear regression) and peak-to-TSS distance had a negative effect (p=2.4 × 10⁻²³, linear regression) on eQTL $h^2_{cis}$ (**Figure 3B**). Thus, for regulatory variants that showed both caQTL and eQTL signals, those with larger effects on chromatin accessibility tended to exhibit larger effects on gene expression, but their eQTL effects diminished with increasing distance from TSSs.

Next, we investigated how caQTL-mediated IMD heritability is distributed with respect to TSS. If caQTLs beyond the typical *cis*-eQTL window of 1 megabase (Mb) around the genes' TSS explain some proportion of IMD heritability, then *cis*-eQTL analyses might require a wider window to detect disease-associated eQTLs. Across the seven diseases, caQTLs within 500 kb of the TSS of expressed genes explained almost all of the caQTL-mediated IMD heritability (92–100%; **Figure 3C**), indicating that regulatory variants are most likely within 500 kb of the target gene's TSS and supporting the use of a 1 Mb window for *cis*-eQTL analyses. Depending on the disease, 41–66% of the caQTL-mediated IMD heritability was detected in distal peaks further than 10 kb from the TSS of expressed genes, further supporting the analysis of regulatory variants beyond promoter regions.

In sum, the power to detect eQTLs diminishes with increasing distance of the variant from the TSS, but the power to detect caQTLs is largely invariant regardless of peak-to-TSS distance (**Figure 3D**). Since $h^2_{med;\ caQTL}$ are distributed mostly within 500 kb of genes' TSS, the IMD loci colocalizing only with caQTLs could still be weak, undetected eQTLs. Under this model, we predicted that increasing the power to detect eQTLs in LCLs, such as increasing sample size, may lead to further eQTL colocalizations in loci in which we observed only caQTL colocalization.

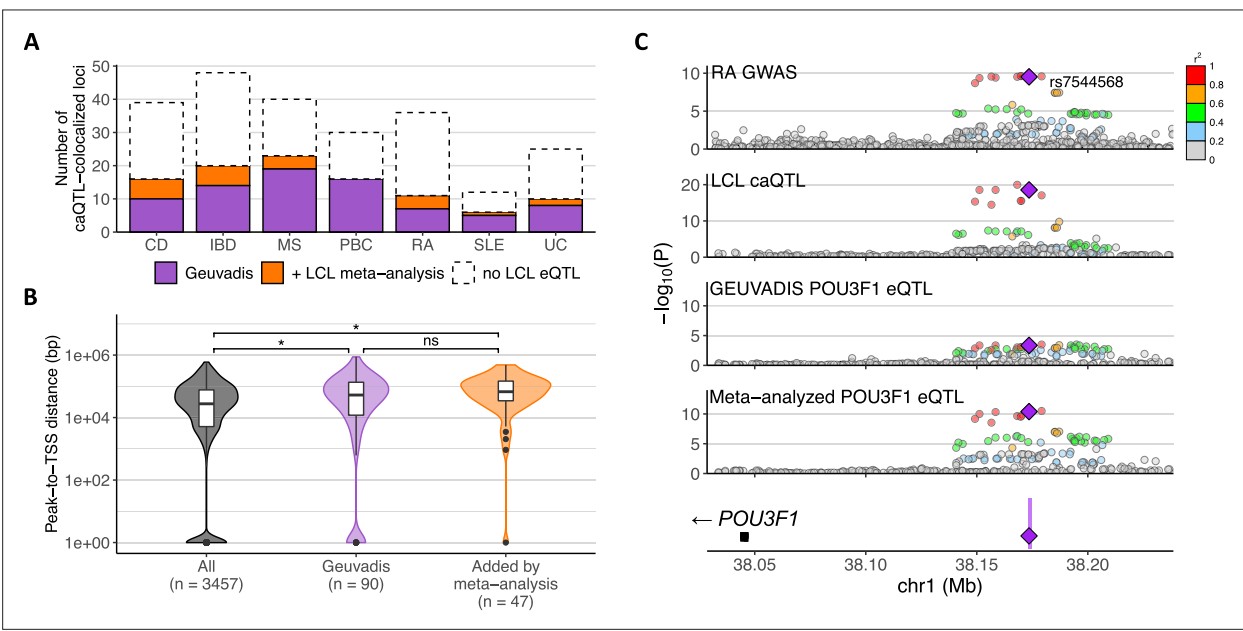

**Figure 4.** Additional colocalization of chromatin accessibility quantitative trait locus (caQTL)-colocalized immune-mediated disease (IMD) loci with meta-analyzed lymphoblastoid cell line (LCL) expression quantitative trait locus (eQTL) data. (**A**) Number of caQTL-colocalized IMD loci that showed eQTL colocalization in LCLs. Disease abbreviations along the *x*-axis are as in **Figure 1**. (**B**) Distribution of peak-to-TSS distance of all caQTL-eQTL pairs and of those colocalized with IMD association. The number of loci in each category is shown in parentheses. *: p<0.01, ns: p>0.05. (**C**) *POU3F1* eQTL that became significantly colocalized with rheumatoid arthritis (RA) association by meta-analyzing LCL eQTL data. Purple shading in the gene plot at the bottom indicates the caQTL peak, and the purple diamond is the lead variant (rs60600003) that is within that peak. The other variants are colored by the degree of linkage disequilibrium (LD) with the annotated variant.

The online version of this article includes the following figure supplement(s) for figure 4:

**Figure supplement 1.** Immune-mediated disease (IMD) genome-wide association study (GWAS) colocalization with meta-analyzed lymphoblastoid cell line (LCL) expression quantitative trait loci (eQTLs).

**Figure supplement 2.** Chromatin accessibility and histone mark levels at immune-mediated disease (IMD)-associated lymphoblastoid cell line (LCL) chromatin accessibility quantitative trait loci (caQTLs).

## Increasing the sample size reveals some eQTL colocalization

For genetic association studies, increasing the sample size is a way to increase statistical power. Therefore, we meta-analyzed four LCL eQTL summary statistics (*Aguet et al., 2020*; *Lappalainen et al., 2013*; *Gutierrez-Arcelus et al., 2013*; *Buil et al., 2015*), leading to a total sample size of 1128 individuals. We performed colocalization analysis using the meta-analyzed summary statistics to evaluate whether effectively increasing the sample sizes would uncover more disease-associated eQTLs in LCLs, especially in loci where a caQTL already showed IMD colocalization. Up to six additional loci showing eQTL colocalization were thus detected for each IMD (*Figure 4A* and *Supplementary file 1D*). For example, *CIITA* is the class II major histocompatibility complex transactivator, which causes severe immunodeficiency if dysfunctional (*Dziembowska et al., 2002*). The *CIITA* locus is associated with IBD right below the genome-wide significance level (rs10445003, p=7.5 × 10$^{-8}$), and it colocalized with a caQTL signal, but initially not with any eQTL (*Figure 4—figure supplement 1A*). However, the meta-analyzed statistics showed a stronger association to *CIITA* expression (p=6.7 × 10$^{-8}$) than without meta-analysis (p=5.4 × 10$^{-4}$) and exhibited a significant colocalization. Interestingly, two of the causal CD genes that previously lacked colocalized eQTLs (*Connally et al., 2022*), *CARD9* and *ATG16L1*, showed significant colocalization in the meta-analyzed LCL eQTL data (*Figure 4—figure supplement 1B and C*).

We hypothesized that increased sample size would improve the power to detect weaker and distal eQTL colocalization. Comparison of the accessibility peak's distance to the paired eQTL gene's (eGene) TSS showed that the newly detected eQTLs tended to be more distal (p=0.06, one-sided Wilcoxon rank-sum test; *Figure 4B*). However, compared to the distribution of the peak-to-TSS distance for all caQTL-eQTL pairs showing colocalization, IMD-associated loci that showed caQTL and eQTL colocalization had greater peak-to-TSS distance on average (p=0.002 for Geuvadis and p=5.9 × 10$^{-6}$ for the meta-analyzed data; *Figure 4B*). For example, an RA-associated locus near *POU3F1*, a neuronal transcription factor that is also induced by interferon (*Hofmann et al., 2010*), colocalized with a distal eQTL located about 126 kb upstream of its promoter, after meta-analysis strengthened the eQTL association (p<10$^{-10}$; *Figure 4C*). These results suggest that additional IMD-associated loci with distal, weaker eQTLs in LCLs might be found if eQTL data were generated for a larger number of individuals.

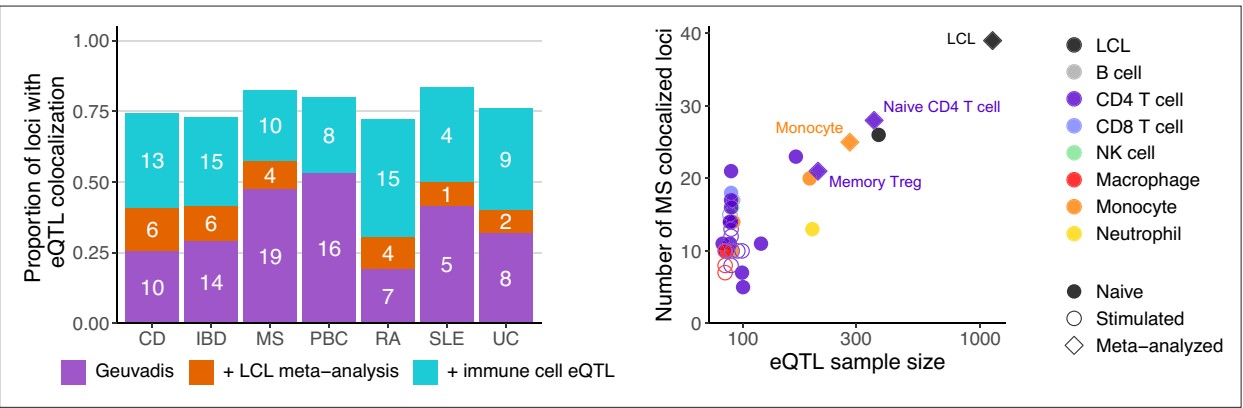

**Figure 5.** Added utility of various immune cell expression quantitative trait locus (eQTL) data. (**A**) Number of loci that additionally colocalized with eQTLs by lymphoblastoid cell line (LCL) meta-analysis (orange) and immune cell data (cyan) compared to the original analysis with Geuvadis LCL eQTL data (purple). The height of the bar is the proportion of loci with eQTL colocalization out of the total immune-mediated disease (IMD) loci with chromatin accessibility quantitative trait locus (caQTL) colocalization in the earlier analysis. Disease abbreviations along the x-axis are as in *Figure 1*. (**B**) Relationship between the number of multiple sclerosis (MS) genome-wide association study (GWAS) loci with eQTL colocalization and sample size for each eQTL dataset. Meta-analyzed eQTL data are labeled with their cell types. NK cell, natural killer cell; Treg, regulatory T cell.

The online version of this article includes the following figure supplement(s) for figure 5:

**Figure supplement 1.** Chromatin accessibility and H3K27ac mark levels at immune-mediated diseases (IMD)-associated chromatin accessibility quantitative trait loci (caQTLs) with respect to monocyte expression quantitative trait locus (eQTL) colocalization.

## IMD loci that colocalized with caQTLs but not eQTLs showed lower levels of active histone marks

Despite uncovering more eQTL colocalizations through meta-analysis, more than 40% of the caQTL-colocalized loci nevertheless showed no colocalization with an eQTL in LCLs (*Figure 4B*). We investigated whether these 'caQTL-only' peaks might be inactive regulatory elements. We quantified the active histone mark levels for H3K27ac, H3K4me1, and H3K4me3 at colocalized caQTL peaks in LCLs and then compared their levels between the 'caQTL and eQTL' and 'caQTL only' loci. On average, H3K27ac marks were stronger at 'caQTL and eQTL' peaks, supporting that the corresponding regulatory elements might be more active (*Figure 4—figure supplement 2*). H3K4me3 marks were detected more often at 'caQTL and eQTL' peaks, leading to stronger average signal. In contrast, H3K4me1 levels were highly similar between the two sets of peaks. Although 'caQTL-only' peaks generally showed lower levels of active histone marks, several individual 'caQTL-only' peaks showed comparable levels. These peaks could be inactive *cis*-regulatory elements in LCLs that affect gene expression in a different cellular context. Therefore, we next examined whether those caQTLs appear as eQTLs in other immune cell types.

## Various immune cell types exhibit eQTL colocalization, where LCLs did not

We downloaded eQTL summary statistics generated from 26 naïve and stimulated immune cell types (*Schmiedel et al., 2018*; *Alasoo et al., 2018*; *Chen et al., 2016*; *Soskic et al., 2022*; *Bossini-Castillo et al., 2022*) to search for eQTLs that may correspond to the remaining, IMD-colocalized caQTLs (*Supplementary file 1E*). The profiled cell types range from B cells and monocytes to subtypes of T cells, as well as stimulated T cells and macrophages. We tested for colocalization of these eQTLs to IMD associations.

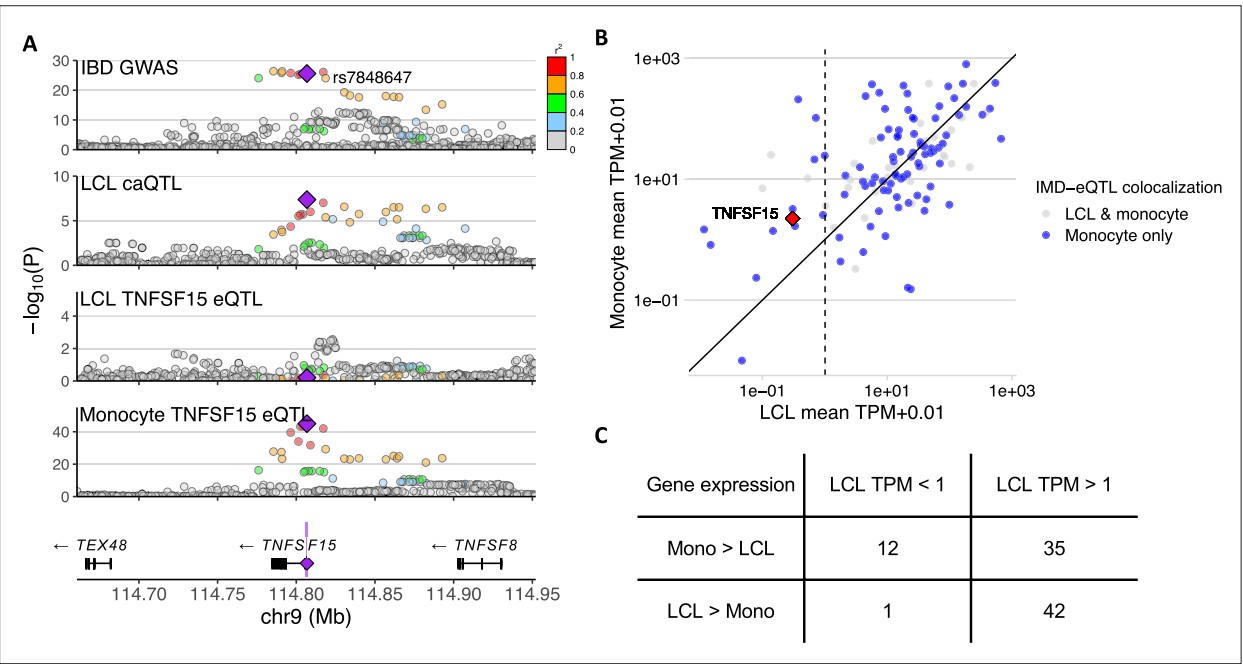

**Figure 6.** Immune-mediated disease (IMD) loci that colocalized with an expression quantitative trait locus (eQTL) in monocytes, but not in lymphoblastoid cell lines (LCLs), even though they colocalized with chromatin accessibility quantitative trait loci (caQTLs) in LCLs. (**A**) *TNFSF15* locus plot showing genetic association to inflammatory bowel disease (IBD), chromatin accessibility in LCLs, and *TNFSF15* expression in LCLs and monocytes. Purple shading in the gene plot at the bottom indicates the caQTL peak, and the purple diamond shows a strongly associated variant (rs7848647) that is within that peak. The other variants are colored by the degree of linkage disequilibrium (LD) with the annotated variant. (**B**) Expression levels of genes in LCLs and monocytes colored according to their eQTL colocalization outcome. TPM: transcripts per million. (**C**) Number of genes with eQTL colocalization in monocytes, but not in LCLs, separated by the gene's expression level in LCLs (column) and whether it is lower or higher than that in monocytes (row).

25–42% of the caQTL-colocalized IMD loci that were missing eQTL colocalizations in LCLs showed eQTL colocalizations in at least one immune cell type (*Figure 5A* and *Supplementary file 1F*). The overlap of LCL caQTLs with non-LCL immune cell eQTLs was greater than expected by chance for 5 of the 7 IMDs (p<0.00714 [Bonferroni-corrected threshold], Fisher's exact test, for CD, IBD, PBC, RA, and UC; *Supplementary file 1G*), suggesting that the caQTLs found in LCLs may also show regulatory function in those immune cell types. Comparing across the datasets, we found that the number of loci with eQTL colocalization varied depending on the cell type, but that the effect of the sample size was more profound ($r^2$=0.60–0.79; *Figure 5B* and *Supplementary file 1H*). We meta-analyzed eQTL data of three immune cell types with multiple sources – naïve $CD4^+$ T cell, monocyte, and memory regulatory T cell (Treg) – and this also increased the number of loci with significant eQTL colocalization (*Supplementary file 1I*). Altogether, these results suggest that although generating eQTL data in more cell types and cell states uncovers context-specific eQTLs, increasing the sample size should also be a priority to ensure sufficient statistical power.

We investigated the potential reasons why IMD loci that colocalized with caQTLs in LCLs showed eQTLs not in LCLs but in other immune cells. First, LCL caQTLs may correspond to gene regulatory elements that exert their effects on gene expression in a different cellular context. For instance, monocyte H3K27ac levels in the 'caQTL-only' loci where monocyte eQTLs colocalized were higher than those with no monocyte eQTLs (*Figure 5—figure supplement 1*). Second, some examples, such as for *TNFSF15*, were due to cell-type-specific gene expression (*Figure 6*): despite a significant colocalization of IBD with a caQTL in LCLs in the *TNFSF15* locus, disease-associated eQTL signal was detected only in monocytes (*Figure 6A*). Tumor necrosis factor-like cytokine 1A (TL1A), the protein encoded by *TNFSF15*, is secreted by monocytes and many other cells to activate helper T cells, Treg, and B cells (*Xu et al., 2022*). *TNFSF15* expression was low in LCLs (mean transcript per million [TPM]=0.30), but higher in monocytes (mean TPM = 2.23). Of the genes that showed exclusively monocyte eQTL colocalization, those with low expression (mean TPM <1) in LCLs generally showed higher expression in monocytes (*Figure 6B and C*). However, low expression in LCLs was likely not the explanation for most cases of 'monocyte-only' eQTLs (blue points in *Figure 6B*) because most 'monocyte-only' eQTL genes were expressed at a level higher than 1 TPM in LCLs.

Overall, expanding the eQTL search to various immune cell types increased the number of eQTL-colocalized loci among those that previously colocalized only with caQTLs (*Figure 5A*). On average, approximately 75% of the caQTL-colocalized loci ultimately showed eQTL colocalization. These results highlight the utility of eQTL data across a range of immune cell types for discovery of IMD-associated eQTLs.

## Discussion

A lack of the link between many noncoding GWAS loci to the associated variants' gene regulatory effects has posed challenges in understanding their genetic mechanism (*Connally et al., 2022*). There have been various hypotheses presented from disease genes showing more complex gene regulation (*Wang and Goldstein, 2020*), context specificity of gene regulation (*Umans et al., 2021*), and a combination of both, due to selective constraints against damaging eQTLs (*Mostafavi et al., 2023*). A better evaluation of the potential reasons for the missing regulation will guide future data generation projects to elucidate disease-associated loci. To determine why some loci might lack colocalized eQTLs, we focused on chromatin accessibility, which is a molecular phenotype affected by regulatory variants more directly than are eQTLs. We approached this question with mediated heritability analysis (*Yao et al., 2020*) and colocalization analysis (*Giambartolomei et al., 2014*; *Pickrell et al., 2016*).

We found that caQTLs in LCLs mediate a significant proportion of heritability for many autoimmune diseases. In contrast, LCLs did not appear to be an effective cell type to model gene regulatory effects in allergic diseases. The $h^2_{med}/h^2_{SNP}$ estimates for caQTLs were higher than those of eQTLs in most autoimmune diseases, even though the smaller sample size of caQTL data (i.e. 100 vs 373) could bias caQTLs' estimates toward zero (*Yao et al., 2020*). We also showed that disease-associated chromatin accessibility effects often share the genetic signal with gene expression effects but that there are also many loci without an eQTL detected in LCLs.

By focusing on disease-associated caQTL that lacked significant eQTLs, we explored how additional colocalized eQTL effects could be uncovered. First, increasing the sample sizes for the eQTL statistics via meta-analysis demonstrated that more eQTL colocalizations can be detected with increased

statistical power and robustness (*Hukku et al., 2021*). These results are consistent with the hypothesis that disease-associated eQTLs are typically weaker and distal due to negative selection against large expression changes for causal genes (*Mostafavi et al., 2023*). Second, many caQTLs in LCLs without eQTLs in LCLs showed eQTL colocalization in other immune cell types. Context specificity of eQTLs has been widely considered to be the primary explanation for the difficulty of pinpointing disease-associated eQTLs (*Umans et al., 2021*; *Schmiedel et al., 2018*; *Alasoo et al., 2018*; *Soskic et al., 2022*). Our observation that many IMD-colocalized caQTLs in LCLs show eQTLs in other immune cell types suggests that caQTL effects may be shared across cell types, whereas eQTL effects are more context-specific (*Alasoo et al., 2018*). If a shared set of transcription factors is expressed in similar yet distinct cell types, like the immune cells, genetic variants affecting their DNA binding would affect chromatin accessibility similarly. On the other hand, regulation of gene expression is a result of multiple regulatory elements, each binding multiple transcription factors (*Kim and Wysocka, 2023*), so the measured effect of a regulatory variant on gene expression likely depends much more on the cellular context. In such cases, eQTL data for the specific cellular contexts need to be generated in future studies to uncover genetic signal shared with a complex trait or disease. All in all, our results suggest that both increasing the sample size and generating gene expression data from more relevant cellular contexts would be useful strategies for discovering more disease-associated eQTLs.

Finally, we demonstrated that caQTLs can reveal the regulatory variant effect of disease-associated variants that may have been difficult to detect with eQTLs, particularly in TSS-distal regions. Although caQTLs cannot directly identify the target gene or the causal cellular context, we anticipate that integrated analyses can improve the power to detect weaker eQTL signals, as multi-trait GWAS analyses have shown (*Turley et al., 2018*). Moreover, integrating multiple molecular QTL data, like transcription factor bQTLs and histone mark QTLs, may highlight the regulatory elements associated with the GWAS phenotype, which may ultimately contribute to identifying the causal gene (*Jeong and Bulyk, 2023*). Therefore, we anticipate the generation of data across the various molecular phenotypes upstream of gene expression for QTL analyses will be informative.

## Limitations of the study

The $h^2_{med}/h^2_{SNP}$ estimates can be biased because of insufficient sample size of the QTL data. Thus, the proportion of mediated IMD heritability could change based on the specific caQTL and eQTL data. Colocalization analysis tests whether the QTL and GWAS data likely share genetic signal, and such shared signal could arise from either causal mediation or pleiotropy. Therefore, further experiments are needed to establish causality of colocalized eQTL genes. Lastly, we hypothesized that caQTL effects are often shared across cell types, but we did not have access to relevant data to test this hypothesis.

## Methods
### ATAC-seq data processing and peak calling

We downloaded LCL ATAC-seq data of British (GBR) samples (n=100) from European Nucleotide Archive (ENA) under accession ERP110508 (*Kumasaka et al., 2019*). The available files were cram alignment files mapped to the b37 reference genome, so we extracted unique read pairs using SAMtools (*Danecek et al., 2021*) and bamtofastq command from bedtools (*Quinlan and Hall, 2010*). The reads were paired-end and each 75 base pairs (bp) long. The data contained reads with Nextera transposase adapters, so we removed the adapter sequences and bases of poor quality at the 3' end using cutadapt. Trimmed reads with both pairs shorter than 20 bp were discarded. The command was 'cutadapt -a file:${forward} -A file:${reverse} -e 0.25 j 2 -q 15 `--pair-filter=both` -m 20'. (${forward} and ${reverse} files contain the forward and reverse Nextera transposase adapters). We mapped the reads to the GRCh38 reference genome using Bowtie 2 (*Langmead and Salzberg, 2012*) with the 'GRCh38_noalt_decoy_as' index provided on the tool's website. The command was 'bowtie2 `--very-sensitive --no-mixed --no-discordant` -I 20 -X 2000'. We kept only read alignments with mapping quality greater than 1. We also removed reads aligning to the mitochondrial genome, those overlapping ENCODE exclusion regions (file ID: ENCFF356LFX) (*Dunham, 2012*) and potential PCR duplicates using scripts from WASP (*van de Geijn et al., 2015*).

To represent peaks across the samples, we subsampled 3 million read pairs from each and pooled them. Then, we used MACS2 (*Zhang et al., 2008*) with the BAMPE option for peak calling. The command was 'macs2 callpeak -f BAMPE -g hs -q 0.05'. We further used the 'bdgcmp' and 'bdgpeakcall' subcommand to find peaks that are at least 100 bp long (-l) and merge those that are less than 100 bp apart (-g). We also merged peaks similarly derived from individual samples using the 'merge' command in bedtools.

Furthermore, for the sake of comprehensiveness, we repeated these steps with LCL ATAC-seq data of Yoruban (YRI) samples (*Banovich et al., 2018*) and merged the peaks with the earlier peak set derived from GBR samples. In total, there were 443,403 peaks genome-wide.

## RNA expression data preparation

We downloaded RNA expression level data of the LCL samples (*Lappalainen et al., 2013*) from the Expression Atlas (*Papatheodorou et al., 2020*). This data consisted of TPM values of genes as processed by the Expression Atlas. We retained TPM values of protein-coding and long noncoding RNAs in European samples for downstream analyses, including QTL analysis, mediated heritability analysis, and colocalization.

## LCL samples' genotype preparation

To utilize the genotype calls of the highest quality, we downloaded the high-coverage 1000 Genomes (1kG) Project data (*Byrska-Bishop et al., 2022*). Of the European samples with ATAC-seq or RNA-seq data, 14 samples had genotypes derived from microarrays (*Auton et al., 2015*), and the remaining samples had genotypes derived from the high-coverage whole-genome sequencing data. The samples with only microarray-based genotypes that needed imputation are listed in *Supplementary file 1J*. We lifted over the microarray data based on the hg19 reference genome to GRCh38 and filtered for variants present in the high-coverage 1kG data. We first imputed microarray genotype data using the TOPMed imputation server (*Das et al., 2016*) and extracted SNPs with imputation $R^2 \geq 0.5$ and imputed the rest of the variants, including short indels, using high-coverage data with Beagle5.2 (*Browning et al., 2018*; *Browning et al., 2021*). After keeping only variants with DR2 $\geq 0.7$, we merged the imputed genotypes with the high-coverage 1kG genotypes.

## Curation of IMD GWAS data

We downloaded GWAS summary statistics for 13 IMDs, including 11 autoimmune diseases (ATD [*Cordell et al., 2021*], CEL [*Dubois et al., 2010*], CD [*de Lange et al., 2017*], IBD [*de Lange et al., 2017*], JIA [*López-Isac et al., 2021*], MS [*Patsopoulos, 2019*], PBC [*Cordell et al., 2021*], RA [*Ishigaki et al., 2022*], SLE [*Bentham et al., 2015*], UC [*de Lange et al., 2017*], and VIT [*Jin et al., 2016*]) and 2 allergic diseases (ALL [*Loh et al., 2018*] and AST [*Loh et al., 2018*]). For each disease, we searched for more recent studies with larger sample sizes and prioritized those with genome-wide statistics, rather than those with only Immunochip variants (*Cortes and Brown, 2011*). To compare with nonimmune diseases, we also downloaded summary statistics for 3 nonimmune diseases (T2D [*Morris et al., 2012*], CAD [*Schunkert et al., 2011*], and SCZ [*Schizophrenia Working Group of the Psychiatric Genomics Consortium, 2014*]). In this study, we analyzed only those GWAS summary statistics derived from cohorts of individuals with European ancestries.

Since the LCL samples' genotypes are based on the GRCh38 reference genome, we lifted over any GWAS data based on b37 reference genome to the GRCh38 genomic coordinates. Briefly, we formatted the summary statistics as bed files and used the liftOver tool (*Kent et al., 2002*) to convert them to GRCh38 genomic coordinates. Then, to ensure that the reference alleles match the sequences of the GRCh38 reference genome, we used the gwas2vcf tool (*Lyon et al., 2021*).

## caQTLs in LCLs

First, we quantified the chromatin accessibility levels at ATAC-seq peaks identified earlier. We counted the number of read fragments overlapping each peak using featureCounts (*Liao et al., 2014*). For each sample, the read counts were normalized for library size using trimmed mean of M-values (*Robinson and Oshlack, 2010*) so that the values are comparable across the samples. Then, the phenotype values were further normalized to follow a standard normal distribution across the samples, using

quantile normalization. Peaks with counts per million (CPM)<0.8 or counts <10 for more than 20% of the samples were discarded.

Next, we performed a principal component analysis (PCA) on the phenotype matrix to derive potential latent covariates. We selected the number of principal components (PCs) to incorporate in the regression model based on the Buja and Eyuboglu algorithm (*Buja and Eyuboglu, 1992*) that is implemented in PCAForQTL (*Zhou et al., 2022*). Ultimately, we accounted for sex, library size, 3 genotype PCs, and 13 phenotype PCs in the QTL analysis. We performed genetic association tests on variants within 200 kilobases (kb) of the peak using tensorQTL (*Taylor-Weiner et al., 2019*). We discarded variants with minor allele frequency less than 5%.

## IMD heritability enrichment in accessible regions of LCLs

We evaluated the relevance of accessible regions in LCLs to IMD heritability using S-LDSC (*Finucane et al., 2015*). We used the baselineLD v2.2 annotation in hg38 and the European LD reference from the 1000 Genomes Project (downloaded from the S-LDSC website, https://alkesgroup.broadinstitute.org/LDSCORE/GRCh38/). We used the set of filtered ATAC-seq peaks that we tested for QTL associations. We accounted for 16 diseases (13 IMDs and 3 non-IMDs) for the Bonferroni-corrected p-value threshold.

## Mediated heritability estimation of QTLs

We estimated the heritability mediated by QTLs ($h^2_{med}$) using MESC (*Yao et al., 2020*). We denote heritability of tested SNPs as $h^2_{SNP}$.

### caQTL-mediated heritability

We estimated the 'expression scores' for chromatin accessibility in LCLs using individual genotypes and phenotypes. We analyzed the same set of peaks and accounted for the same covariates as we did for the QTL analysis above. For mediated heritability estimation, we accounted for baseline LD v2.2 annotation in hg38 without the QTL annotations, as they could be redundant. The estimand of interest is the proportion of heritability mediated by caQTLs ($h^2_{med}/h^2_{SNP}$).

To evaluate whether certain peak sets are enriched for mediated heritability, we utilized the gene set analysis functionality. For peak strength, we considered the 95% percentile CPM value of each peak and stratified the peaks into quintiles. For histone marks, we first generated histone ChIP-seq peaks using data from *Delaneau et al., 2019*. Similar to calling ATAC-seq peaks, we sampled 3 million reads per sample and merged them before applying MACS2 (*Zhang et al., 2008*). We downloaded three control ChIP-seq data from ENCODE to use as input (File IDs: ENCFF066RCS, ENCFF159XTB, and ENCFF850RIE) (*Dunham, 2012*). Then, we curated sets of ATAC-seq peaks that overlapped H3K27ac, H3K4me1, and H3K4me3 ChIP-seq peaks by at least 1 bp. ATAC-seq peaks that overlapped H3K27ac but not H3K4me3 regions were labeled as 'enhancer-like signature', while those that overlapped H3K27ac and H3K4me3 regions were labeled as 'promoter-like signature'. The estimand of interest is the proportion of mediated heritability explained by the peak set (peak set $h^2_{med}$/total $h^2_{med}$).

### Comparison with eQTL-mediated heritability

First, we estimated $h^2_{med}/h^2_{SNP}$ of eQTLs (i.e. $h^2_{med; eQTL}/h^2_{SNP}$) the same way as we did for that of caQTLs. Then, we also estimated $h^2_{med}/h^2_{SNP}$ of caQTLs and eQTLs together (i.e. $h^2_{med; caQTL \cup eQTL}$) with MESC meta-analysis (*Yao et al., 2020*). caQTLs and eQTLs were also stratified as separate sets to account for potential differences in the relationship of QTL *cis*-heritability and $h^2_{med}$. This meta-analyzed estimate is effectively the amount of heritability mediated by either caQTLs or eQTLs in LCLs. This estimate reveals the overall relationship between heritability mediated by caQTLs and eQTLs. For instance, the estimate of the heritability mediated exclusively by caQTLs would be $h^2_{med; just caQTL} = h^2_{med; caQTL \cup eQTL} - h^2_{med; caQTL}$. The estimate of mediated heritability shared by caQTLs and eQTLs is $h^2_{med; caQTL \cap eQTL} = (h^2_{med; caQTL} + h^2_{med; eQTL}) - h^2_{med; caQTL \cup eQTL}$.

## Colocalization analyses
### caQTL and eQTL colocalization with IMD GWAS

First, we selected candidate colocalization loci by filtering for overlapping 'significant' variants. The candidate loci met the following conditions: (1) lead IMD association at p<10⁻⁶, (2) lead caQTL or

eQTL association at $p<10^{-4}$, and (3) at least one variant simultaneously showed caQTL or eQTL $\chi^2$ statistics greater than 0.8×lead $\chi^2$ statistics for the caQTL or eQTL and IMD association $\chi^2$ statistics greater than $0.8 \times \chi^2$ statistics for the IMD lead variant in the locus. Then, we applied gwas-pw (*Pickrell et al., 2016*) on the variants within 100 kb of the lead variant. We considered loci with posterior probability of colocalization (PP3)>0.98 to be colocalized (*Kundu et al., 2022*).

## Colocalization of caQTL with eQTL

We performed a colocalization analysis of caQTLs and eQTLs to curate a set of loci where the same genetic signal likely explains both associations. The pairs of caQTLs and eQTLs reveal the distance between the regulatory element and the target gene's TSS. We selected candidate colocalization loci with: (1) IMD association at $p<10^{-6}$, (2) QTL association at $p<10^{-4}$, and (3) the two lead variants showed LD $r^2>0.8$. We applied gwas-pw (*Pickrell et al., 2016*) on the variants within 200 kb of the tested caQTL peak. We considered loci with posterior probability of colocalization (PP3)>0.98 to be colocalized (*Kundu et al., 2022*).

## Overlap of protein factor ChIP-seq and colocalized caQTL peaks

We searched for any protein factors detected at the colocalized caQTL peaks using the Cistrome database (*Liu et al., 2011*), which we accessed on July 24, 2024. We considered only ChIP-seq data from immune cell types, progenitors, and stem cells that can differentiate into immune cells. For each ChIP-seq peak, we searched for those that show overlap of more than 50% with one of 305 caQTL peaks that colocalized with IMD GWAS signals (*Supplementary file 1A*). Each protein factor detected in the caQTL peaks and the cell type used to generate the ChIP-seq data are listed in *Supplementary file 1C*.

## Enrichment of biological processes

To test whether colocalized genes are likely relevant to autoimmune diseases, we surveyed which Biological Process GO terms were overrepresented compared to all the genes within 500 kb of each IMD association signal. Enrichment of biological processes was evaluated using Protein Analysis Through Evolutionary Relationships (PANTHER) (*Mi et al., 2019*). The foreground list comprised all of the genes whose eQTL signal colocalized with one of the seven autoimmune diseases (CD, IBD, MS, PBC, RA, SLE, and UC). The background list of genes was all of the genes within 500 kb of each IMD lead variant for which we observed colocalization. Moreover, we tested whether colocalized caQTLs without eQTLs are closer to genes related to immune processes than expected by chance. For this, we used Genomic Regions Enrichment of Annotations Tool (GREAT) (*McLean et al., 2010*). The foreground list comprised ATAC-seq peaks at IMD loci showing colocalization only with caQTLs and not with eQTLs. The background list is all ATAC-seq peaks identified in LCLs that we tested for caQTL association.

## Relationship between peak-to-TSS distance and *cis*-heritability of caQTL and eQTL

The distance between the colocalized caQTL peak and the eGene's TSS (i.e. peak-to-TSS distance) was determined to be the shortest distance from one end of the caQTL peak to the TSS. The pairs of caQTLs and eQTLs were split into quintiles based on their peak-to-TSS distance from closest to farthest.

MESC analysis uses REML implemented in GCTA (*Yang et al., 2011*) to estimate QTL *cis*-heritability. We referred to its output and compared the *cis*-heritability of caQTLs and eQTLs based on peak-to-TSS distance. To visualize the distribution of *cis*-heritability estimates, we grouped pairs of colocalized caQTLs and eQTLs based on peak-to-TSS distance quintiles.

## LCL eQTL meta-analysis

We downloaded LCL eQTL data from three studies through eQTL Catalogue release 6 (*Kerimov et al., 2021*). The sample sizes were 190, 147, and 418 for GENCORD (*Gutierrez-Arcelus et al., 2013*), GTEx (*Aguet et al., 2020*), and TwinsUK (*Buil et al., 2015*), respectively. We meta-analyzed the summary statistics using the inverse variance weighted fixed effects model. If a variant was missing

in a subset of the studies, then only the available statistics were meta-analyzed. We used these meta-analyzed statistics to perform colocalization the same way as earlier colocalization analyses.

### Colocalization analysis with immune cell eQTL data

We downloaded eQTL data for various immune cell types from the eQTL Catalogue release 6 (*Kerimov et al., 2021*). The source studies from the eQTL Catalogue include BLUEPRINT (*Chen et al., 2016*), DICE (*Schmiedel et al., 2018*), *Alasoo et al., 2018*, and *Bossini-Castillo et al., 2022*. The represented immune cell types include T cell subtypes, B cells, neutrophils, and monocytes. We also separately downloaded data for CD4$^+$ T cells with and without stimulation (*Soskic et al., 2022*). The selection of candidate loci and colocalization analysis on them followed the same procedure as that for other QTLs.

We also evaluated whether meta-analysis of eQTL data can increase the number of loci with eQTL colocalization. We meta-analyzed eQTL data for three immune cell types with multiple sources – naïve CD4$^+$ T cell, monocyte, and memory Treg – using the inverse variance weighted fixed effects model. Specifically, we meta-analyzed naïve CD4$^+$ T cell eQTL summary statistics from *Soskic et al., 2022*, BLUEPRINT (*Chen et al., 2016*), and DICE (*Schmiedel et al., 2018*). We meta-analyzed monocyte eQTL summary statistics from BLUEPRINT (*Chen et al., 2016*) and DICE (*Schmiedel et al., 2018*). We meta-analyzed memory Treg eQTL summary statistics from *Bossini-Castillo et al., 2022*, and DICE (*Schmiedel et al., 2018*). We used these meta-analyzed statistics to perform colocalization the same way as earlier colocalization analyses.

## Acknowledgements

We thank Vijay Sankaran, Shamil Sunyaev, Alexander Gusev, and members of the Raychaudhuri lab for helpful discussion. This work was funded by NIH grant R01 HG010501.

## Additional information

### Funding

| Funder | Grant reference number | Author |
| --- | --- | --- |
| National Human Genome Research Institute | R01 HG010501 | Martha L Bulyk |

The funders had no role in study design, data collection and interpretation, or the decision to submit the work for publication.

### Author contributions

Raehoon Jeong, Conceptualization, Data curation, Software, Formal analysis, Investigation, Visualization, Methodology, Writing – original draft, Writing – review and editing; Martha L Bulyk, Conceptualization, Resources, Supervision, Funding acquisition, Methodology, Project administration, Writing – review and editing

### Author ORCIDs

Raehoon Jeong ⓘ https://orcid.org/0000-0001-9840-4692
Martha L Bulyk ⓘ https://orcid.org/0000-0002-3456-4555

Reviewer #1 (Public review): https://doi.org/10.7554/eLife.98289.3.sa1
Reviewer #2 (Public review): https://doi.org/10.7554/eLife.98289.3.sa2
Author response https://doi.org/10.7554/eLife.98289.3.sa3

## Additional files

### Supplementary files

MDAR checklist

Supplementary file 1. Supplementary tables 1A-J.

## Data availability

Processed data and code for generating the figures presented in the manuscript are available at https://github.com/BulykLab/IMD-colocalization-manuscript-figures (copy archived at *Jeong, 2024*).

The following previously published datasets were used:

| Author(s) | Year | Dataset title | Dataset URL | Database and Identifier |
|---|---|---|---|---|
| Byrska-Bishop M, Evani US, Zhao X | 2022 | High-coverage whole-genome sequencing of the expanded 1000 Genomes Project cohort including 602 trios | https://www.internationalgenome.org/data-portal/data-collection/30x-grch38 | The International Genome Sample Resource, 30x-grch38 |
| Kumasaka N, Knights AJ, Gaffney DJ | 2018 | High-resolution genetic mapping of putative causal interactions between regions of open chromatin | https://www.ebi.ac.uk/ena/browser/view/PRJEB28318 | European Nucleotide Archive, PRJEB28318 |
| Lappalainen T, Sammeth M, Friedländer MR | 2013 | Transcriptome and genome sequencing uncovers functional variation in humans | https://www.ebi.ac.uk/gxa/experiments/E-GEUV-1/Results | Expression atlas, E-GEUV-1 |
| Cordell HJ, Fryett JJ, Ueno K | 2021 | An international genome-wide meta-analysis of primary biliary cholangitis: Novel risk loci and candidate drugs | https://www.ebi.ac.uk/gwas/studies/GCST90061440 | GWAS Catalog, GCST90061440 |
| Dubois PC, Trynka G, Franke L | 2010 | Multiple common variants for celiac disease influencing immune gene expression | https://www.ebi.ac.uk/gwas/studies/GCST000612 | GWAS Catalog, GCST000612 |
| de Lange KM, Moutsianas L, Lee JC | 2017 | Genome-wide association study implicates immune activation of multiple integrin genes in inflammatory bowel disease | https://www.ebi.ac.uk/gwas/studies/GCST004131 | GWAS Catalog, GCST004131 |
| López-Isac E, Smith SL, Marion MC | 2021 | Combined genetic analysis of juvenile idiopathic arthritis clinical subtypes identifies novel risk loci, target genes and key regulatory mechanisms | https://www.ebi.ac.uk/gwas/studies/GCST90010715 | GWAS Catalog, GCST90010715 |
| Ishigaki K, Sakaue S, Terao C | 2022 | Multi-ancestry genome-wide association analyses identify novel genetic mechanisms in rheumatoid arthritis | https://www.ebi.ac.uk/gwas/studies/GCST90132223 | GWAS Catalog, GCST90132223 |
| Bentham J, Morris DL, Graham DSC | 2015 | Genetic association analyses implicate aberrant regulation of innate and adaptive immunity genes in the pathogenesis of systemic lupus erythematosus | https://www.ebi.ac.uk/gwas/studies/GCST003156 | GWAS Catalog, GCST003156 |
| Jin Y, Andersen G, Yorgov D | 2016 | Genome-wide association studies of autoimmune vitiligo identify 23 new risk loci and highlight key pathways and regulatory variants | https://www.ebi.ac.uk/gwas/studies/GCST004785 | GWAS Catalog, GCST004785 |

*Continued on next page*

*Continued*

| Author(s) | Year | Dataset title | Dataset URL | Database and Identifier |
|---|---|---|---|---|
| Schmiedel BJ, Singh D, Madrigal A | 2018 | Impact of Genetic Polymorphisms on Human Immune Cell Gene Expression | https://www.ebi.ac.uk/eqtl/Studies/ | eQTL Catalogue, QTS000026 |
| Chen L, Ge B, Casale FP | 2016 | Genetic Drivers of Epigenetic and Transcriptional Variation in Human Immune Cells | https://www.ebi.ac.uk/eqtl/Studies/ | eQTL Catalogue, QTS000002 |
| Bossini-Castillo L, Glinos DA, Kunowska N | 2022 | Immune disease variants modulate gene expression in regulatory CD4+ T cells | https://www.ebi.ac.uk/eqtl/Studies/ | eQTL Catalogue, QTS000003 |
| Soskic B, Cano-Gamez E, Smyth DJ | 2022 | Immune disease risk variants regulate gene expression dynamics during CD4+ T cell activation | https://doi.org/10.5281/zenodo.6006795 | Zenodo, 10.5281/zenodo.6006795 |
| Alasoo K, Rodrigues J, Mukhopadhyay S | 2018 | Shared genetic effects on chromatin and gene expression indicate a role for enhancer priming in immune response | https://www.ebi.ac.uk/eqtl/Studies/ | eQTL Catalogue, QTS000001 |
| de Lange KM, Moutsianas L, Lee JC | 2017 | Genome-wide association study implicates immune activation of multiple integrin genes in inflammatory bowel disease | https://www.ebi.ac.uk/gwas/studies/GCST004132 | GWAS Catalog, GCST004132 |
| de Lange KM, Moutsianas L, Lee JC | 2017 | Genome-wide association study implicates immune activation of multiple integrin genes in inflammatory bowel disease | https://www.ebi.ac.uk/gwas/studies/GCST004133 | GWAS Catalog, GCST004133 |

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
