## [Editor Report · eLife Assessment]

This paper addresses a significant question regarding the low overlap between genetic discoveries for human complex diseases and those for gene expression by emphasizing the contribution of cell-type-specific chromatin accessibility QTLs. The analyses supporting the main claims are **convincing**, and the key conclusions are **valuable** and of interest to readers in the fields of human genetics and functional genomics.

---

## [Referee Report · Reviewer #1 (Public review)]

Most human traits and common diseases are polygenic, influenced by numerous genetic variants across the genome. These variants are typically non-coding and likely function through gene regulatory mechanisms. To identify their target genes, one strategy is to examine if these variants are also found among genetic variants with detectable effects on gene expression levels, known as eQTLs. Surprisingly, this strategy has had limited success, and most disease variants are not identified as eQTLs, a puzzling observation recently referred to as "missing regulation".

In this work, Jeong and Bulyk aimed to better understand the reasons behind the gap between disease-associated variants and eQTLs. They focused on immune-related diseases and used lymphoblastoid cell lines (LCLs) as a surrogate for the cell types mediating the genetic effects. Their main hypothesis is that some variants without eQTL evidence might be identifiable by studying other molecular intermediates along the path from genotype to phenotype. They specifically focused on variants that affect chromatin accessibility, known as caQTLs, as a potential marker of regulatory activity.

The authors present data analyses supporting this hypothesis: several disease-associated variants are explained by caQTLs but not eQTLs. They further show that although caQTLs and eQTLs likely have largely overlapping underlying genetic variants, some variants are discovered only through one of these mapping strategies. Notably, they demonstrate that eQTL mapping is underpowered for gene-distal variants with small effects on gene expression, whereas caQTL mapping is not dependent on the distance to genes. Additionally, for some disease variants with caQTLs but no corresponding eQTLs in LCLs, they identify eQTLs in other cell types.

Altogether, Jeong and Bulyk convincingly demonstrate that for immune-related diseases, discovering the missing disease-eQTLs requires both larger eQTL studies and a broader range of cell types in expression assays. It remains to be seen what fractions of the missing disease-eQTLs will be discovered with either strategy and whether these results can be extended to other diseases or traits.

It should be noted that the problem of "missing regulation" has been investigated and discussed in several recent papers, notably Umans et al., Trends in Genetics 2021; Connally et al., eLife 2022; Mostafavi et al., Nat. Genet. 2023. The results reported by Jeong and Bulyk are not unexpected in light of this previous work (all of which they cite), but they add valuable empirical evidence that mostly aligns with the model and discussions presented in Mostafavi et al.

---

## [Referee Report · Reviewer #2 (Public review)]

eQTLs have emerged as a method for interpreting GWAS signals. However, some GWAS signals are difficult to explain with eQTLs. In this paper, the authors demonstrated that caQTLs can explain these signals. This suggests that for GWAS signals to actually lead to disease phenotypes, they must be accessible in the chromatin. This implies that for GWAS signals to translate into disease phenotypes, they need to be accessible within the chromatin.

However, fundamentally, caQTLs, like GWAS, have the limitation of not being able to determine which genes mediate the influence on disease phenotypes. This limitation is consistent with the constraints observed in this study.

(1) Reproducibility / Methods. The concrete numbers provided in the authors' response (e.g., 20 YRI LCL ATAC‑seq samples used only for peak discovery; caQTL mapping restricted to 100 GBR LCLs; 99,320 ATAC peaks tested vs 14,872 genes for eQTL; 373 European RNA‑seq samples, with clarification of overlap) do not appear to be reflected in the Methods. These specifics should be incorporated directly into the Methods sections.

(2) Experimental evidence demonstrating transcription factor binding at representative caQTL peaks would strengthen causal interpretation of these loci.

(3) Tissue/cell‑type specificity of caQTLs: Prior work supports that chromatin‑level effects are broadly shared across cellular states, whereas expression effects are more context‑specific; thus, caQTLs are generally less "state‑specific" than eQTLs. However, this does not imply equivalence across distinct cell types: caQTLs derived from different cell types may yield different results, particularly where accessibility is cell‑type restricted.

---

## [Author Response]

The following is the authors’ response to the original reviews.

**Reviewer #1 (Public Review):**
Most human traits and common diseases are polygenic, influenced by numerous genetic variants across the genome. These variants are typically non-coding and likely function through gene regulatory mechanisms. To identify their target genes, one strategy is to examine if these variants are also found among genetic variants with detectable effects on gene expression levels, known as eQTLs. Surprisingly, this strategy has had limited success, and most disease variants are not identified as eQTLs, a puzzling observation recently referred to as "missing regulation".In this work, Jeong and Bulyk aimed to better understand the reasons behind the gap between disease-associated variants and eQTLs. They focused on immune-related diseases and used lymphoblastoid cell lines (LCLs) as a surrogate for the cell types mediating the genetic effects. Their main hypothesis is that some variants without eQTL evidence might be identifiable by studying other molecular intermediates along the path from genotype to phenotype. They specifically focused on variants that affect chromatin accessibility, known as caQTLs, as a potential marker of regulatory activity.The authors present data analyses supporting this hypothesis: several disease-associated variants are explained by caQTLs but not eQTLs. They further show that although caQTLs and eQTLs likely have largely overlapping underlying genetic variants, some variants are discovered only through one of these mapping strategies. Notably, they demonstrate that eQTL mapping is underpowered for gene-distal variants with small effects on gene expression, whereas caQTL mapping is not dependent on the distance to genes. Additionally, for some disease variants with caQTLs but no corresponding eQTLs in LCLs, they identify eQTLs in other cell types.Altogether, Jeong and Bulyk convincingly demonstrate that for immune-related diseases, discovering the missing disease-eQTLs requires both larger eQTL studies and a broader range of cell types in expression assays. It remains to be seen what fractions of the missing diseaseeQTLs will be discovered with either strategy and whether these results can be extended to other diseases or traits.

We thank the reviewer for their accurate summary of our study and positive review of our findings for immune-related diseases.

It should be noted that the problem of "missing regulation" has been investigated and discussed in several recent papers, notably Umans et al., Trends in Genetics 2021; Connally et al., eLife 2022; Mostafavi et al., Nat. Genet. 2023. The results reported by Jeong and Bulyk are not unexpected in light of this previous work (all of which they cite), but they add valuable empirical evidence that mostly aligns with the model and discussions presented in Mostafavi et al.

We thank the reviewer for their positive review of our results and manuscript. As Reviewer #1 noted, whether our and others' observation extends to other diseases or traits is an open question. For instance, Figure 2b in Mostafavi et al., Nat. Genet. (2023) demonstrated that there was a spectrum of depletion of eQTLs and enrichment of GWAS signals in constrained genes across various tissues and traits, respectively. Therefore, gene expression constraint may play a larger or smaller role in different diseases or traits. That immune cell types and cell states are extremely diverse (Schmiedel et al., Cell (2018) and Calderon et al., Nat. Genet. (2019), just to name a few) likely adds to the complexity of gene regulation that contributes to immune-mediated disease.

**Reviewer #2 (Public Review):**
Summary:eQTLs have emerged as a method for interpreting GWAS signals. However, some GWAS signals are difficult to explain with eQTLs. In this paper, the authors demonstrated that caQTLs can explain these signals. This suggests that for GWAS signals to actually lead to disease phenotypes, they must be accessible in the chromatin. This implies that for GWAS signals to translate into disease phenotypes, they need to be accessible within the chromatin.However, fundamentally, caQTLs, like GWAS, have the limitation of not being able to determine which genes mediate the influence on disease phenotypes. This limitation is consistent with the constraints observed in this study.

We thank the reviewer for their accurate summary of our results.

(1) For reproducibility, details are necessary in the method section.Details about adding YRI samples in ATAC-seq: For example, how many samples are there, and what is used among public data? There is LCL-derived iPSC and differentiated iPSC (cardiomyocytes) data, not LCL itself. How does this differ from LCL, and what is the rationale for including this data despite the differences?

Banovich et al., Genome Research (2018) (PMID: 29208628), who generated data using LCLderived iPSCs and differentiated iPSCs (cardiomyocytes), also generated ATAC-seq data from 20 YRI LCL samples. We analyzed those data to identify open chromatin regions (i.e., ATACseq peaks) in LCLs and merged the regions with open chromatin regions identified with 100 GBR LCL samples from two studies by Kumasaka et al. Nature Genetics (2016)

PMID: 26656845 and Nature Genetics (2019) PMID: 30478436. However, we restricted the caQTL analysis to only the 100 GBR samples because of possible ancestry effects and batch effects. We attempted caQTL analysis with the 20 YRI samples as well, but the result was noisy, likely due to smaller sample size and lower read depth of the ATAC-seq data.

caQTL is described as having better power than eQTL despite having fewer samples. How does the number of ATAC peaks used in caQTL compare to the number of gene expressions used in eQTL?

The number of ATAC peaks used in caQTL (99,320) is ~6.7 times greater than the number of genes (14,872) used in the eQTL analysis. Therefore, there is a higher chance of detecting a significant caQTL signal and a significant colocalization signal than there is for eQTLs. However, we reasoned that since distal eQTLs are more easily detected as caQTLs and since increasing the sample size of eQTLs through meta-analysis uncovered additional eQTL colocalization at loci with caQTL colocalization only, colocalized caQTLs are likely capturing disease-relevant regulatory effects.

Details about RNA expression data: In the method section, it states that raw data (ERP001942) was accessed, and in data availability, processed data (E-GEUV-1) was used. These need to be consistent.

Thank you for pointing this out. We used the processed data from Expression Atlas (https://www.ebi.ac.uk/gxa/experiments/E-GEUV-1/Results), and that's what we meant by "We downloaded RNA expression level data of the LCL samples from the Expression Atlas." We have revised the “RNA expression data preparation” section in our manuscript to make the text clearer.

How many samples were used (the text states 373, but how was it reduced from the original 465, and the total genotype is said to be 493 samples while ATAC has n=100; what are the 20 others?), and it mentions European samples, but does this exclude YRI?

We thank the reviewer for pointing out these points of confusion. Our reported count of 493 samples included YRI samples with RNA-seq data or ATAC-seq data that we ultimately did not use for QTL analyses. There were 373 European samples with RNA-seq data that we used for eQTL analysis, and 100 GBR samples (including some that overlap with the 373 European samples) that we used for caQTL analysis. We have revised the text to clarify these points.

(2) Experimental results determining which TFs might bind to the representative signals of caQTL are required.

We agree that caQTL colocalization is just the start of elucidating the regulatory mechanism of a GWAS locus. Determining which TFs are bound and which TFs' binding is altered would be necessary to describe the causal regulatory mechanism. For this, we utilized the Cistrome database to search for TFs whose binding overlaps the colocalized caQTL peaks. We present the results of this analysis in Supplementary Table 3 and Supplementary Figure 4, both of which we have added in our revised manuscript. Overall, protein factors associated with active transcription, such as POL2RA, and several immune cell TFs, including RUNX3, SPI1, and RELA, were frequently detected in those peaks. Detecting these factors in most peaks supports the likelihood that the colocalized caQTL peaks are active cis-regulatory elements. These results are consistent with our observation of enriched caQTL-mediated heritability in regions with active histone marks (Figure 1).

(3) It is stated that caQTL is less tissue-specific compared to eQTL; would caQTL performed with ATAC-seq results from different cell types, yield similar results?

We thank the reviewer for the question. Calderon et al. (PMID: 31570894) observed that "most effects on allelic imbalance (of ATAC-seq) were shared regardless of lineage or condition". Yet, there were regions where a different cell type or state would show inaccessibility (Figure 4d in Calderon et al.). Thus, we expect that ATAC-seq results from different cell types (e.g., T cells, B cells, monocytes, etc.) would lead to additional caQTLs showing colocalization at cell-typespecific open chromatin. However, if a region is accessible in both cell types, caQTL may be detected in both. Moreover, Alasoo et al., Nature Genetics (2018) (PMID: 29379200) observed that “many disease-risk variants affect chromatin structure in a broad range of cellular states, but their effects on expression are highly context specific.” In both studies, the authors investigated immune cell types, and there could be different observations in non-immune cell types and other diseases and traits.

**Reviewer #1 (Recommendations For The Authors):**
I think it would strengthen the paper to explore gene-level differences in the discovery of caQTLs and eQTLs. For example, complex disease-relevant genes, on average, have more/longer regulatory domains (as shown by Wang and Goldstein, AJHG 2020; Mostafavi et al., Nat. Genet. 2023). Therefore, it is plausible that for such genes, caQTLs are much more easily discoverable than eQTLs due to (i) a larger mutational target size for caQTLs, and (ii) dispersion of expression heritability across multiple domains, which hampers the discovery of eQTLs but not caQTLs, which are studied independently of other domains in the region. In other words, discovered caQTLs and eQTLs likely vary in terms of their distance to genes (as the authors report), as well as their target genes.

We thank the reviewer for the suggestion to explore gene-level differences. We expect that the effects of complex disease-relevant genes having more / longer regulatory domains, on average, to explain our observations. We agree on both of your points that there are many more regulatory elements that are captured as accessible regions than expressed genes and that genes often have multiple independent eQTLs leading to dispersion of heritability. The genelevel trend that we described was the distance of the regulatory element from the genes. Additional analyses would be a relevant future direction.

Also considering gene-level analysis, Mostafavi et al. show that the types of biases they report for eQTLs also apply to other molecular QTLs. It would be valuable to compare GWAS hits with versus without caQTL colocalization. Similarly, it would be insightful to compare GWAS hits with both colocalized caQTLs and eQTLs to GWAS hits with colocalized caQTLs but no eQTLs in any of the cell types.

We thank the reviewer for the comment. Investigating for potential biases in the colocalized caQTL would be useful, but we considered it beyond the scope of this work. In terms of biological factors, we demonstrated through mediated heritability analyses that more accessible chromatin (based on ATAC-seq read coverage) and regions with active histone marks were enriched for autoimmune disease associations (Figure 1). Furthermore, as greater distance of the regulatory variant from the transcription start site significantly reduced the cis-heritability, we would expect that distance would play a major role, similar to Mostafavi et al.’s conclusions.

I don't think the argument for the role of natural selection contributing to the "missing regulation" is presented accurately. Specifically, large eQTLs acting on top trait-relevant genes are under stronger selection and thus, on average, segregate at lower frequencies. This makes them difficult to discover in eQTL assays. However, if not lost, they contribute as much, if not more, to trait heritability than weaker eQTLs at the same gene because their larger effects compensate for their lower frequency. At the most extreme, selection should have a "flattening" effect (e.g., see Simons et al., PLOS Biol 2018; O'Connor et al., AJHG 2019): weak and strong eQTLs at the same gene are expected to contribute equally to heritability. Therefore, the statement "Consequently, only weak eQTL variants, often in regions distal to the gene's promoter, may remain and affect traits" is not correct. If this turns out to be empirically true, other models, such as pleiotropic selection, need to explain it.

We thank the reviewer for the correction. We agree with the comment and have revised the sentences in the introduction accordingly.

It is worth speculating why caQTLs may be more consistent across cell types than cis-eQTLs. Additionally, readers may infer from the paper that the focus should shift from eQTLs to caQTLs, which may not be the authors' intention. Perhaps these approaches are complementary: caQTLs can help with TSS-distal disease variants, while finding the target gene and regulatory context is more straightforward with eQTL colocalization. Addressing these points in the discussion will be helpful.

We appreciate the reviewer's suggestion to clarify the advantages of incorporating cis-eQTLs and caQTLs. Our argument is exactly as you put it, and we added a paragraph on this in the Discussion.

I believe the authors could do more to contextualize their findings within the existing literature on the subject, particularly Umans et al., Trends in Genetics 2021; Connally et al., eLife 2022; and Mostafavi et al., Nat. Genet. 2023. For instance, Umans et al. suggest that "if most standard eQTLs are generally benign, increasing sample size and adding more tissue types in an effort to identify even more standard eQTLs may not help us to explain many more disease risk mutations". Conversely, Mostafavi et al. argue for a multipronged approach, which appears more aligned with the authors' conclusions.

We followed the reviewer’s suggestion to place our work in the context of existing literature on this topic. Moreover, we clarified what our recommendations for future data generation are.

I thought Figures 1C-D were unclear.

We added a sentence in the figure legend describing that stronger and more significant enrichment indicate that mediated heritability is concentrated in that subset.

**Reviewer #2 (Recommendations For The Authors):**
Complete workflow figures for caQTL calling and eQTL calling are required.

To improve clarity of the caQTL and eQTL calling workflow, we added Supplementary Figure 1**.**